# Incomplete vesicular docking limits synaptic strength under high release probability conditions

Gerardo Malagon[1,2], Takafumi Miki[1,3], Van Tran[1], Laura C Gomez[1], Alain Marty[1]*

[1]Université de Paris, SPPIN-Saints Pères Paris Institute for the Neurosciences, CNRS, Paris, France; [2]Department of Cell Biology and Physiology, Washington University, St. Louis, United States; [3]Graduate School of Brain Science, Doshisha University, Kyoto, Japan

**Abstract** Central mammalian synapses release synaptic vesicles in dedicated structures called docking/release sites. It has been assumed that when voltage-dependent calcium entry is sufficiently large, synaptic output attains a maximum value of one synaptic vesicle per action potential and per site. Here we use deconvolution to count synaptic vesicle output at single sites (mean site number per synapse: 3.6). When increasing calcium entry with tetraethylammonium in 1.5 mM external calcium concentration, we find that synaptic output saturates at 0.22 vesicle per site, not at 1 vesicle per site. Fitting the results with current models of calcium-dependent exocytosis indicates that the 0.22 vesicle limit reflects the probability of docking sites to be occupied by synaptic vesicles at rest, as only docked vesicles can be released. With 3 mM external calcium, the maximum output per site increases to 0.47, indicating an increase in docking site occupancy as a function of external calcium concentration.

## Introduction

The release of synaptic vesicles (SVs) at mammalian central synapses is a complex process that comprises two series of events. Firstly, SVs move to a small part of the presynaptic terminal, the active zone (AZ), where they bind to specific proteins such as RIM and Munc13, and undergo a number of maturation steps including docking and priming (*Südhof, 2012*; *Jahn and Fasshauer, 2012*). At the end of these preparatory processes, SVs are connected to the plasma membrane through SNARE complexes, and they are ready for exocytosis. Secondly, mature SVs fuse with the plasma membrane following a rapid increase in local $Ca^{2+}$ concentration ($Ca_i$), leading to the release of neurotransmitter. The sensitivity of the exocytosis step to $Ca_i$ results from the change of conformation of the SNARE complex following rapid $Ca^{2+}$ binding to a series of $Ca^{2+}$ binding sites residing on synaptotagmins (*Jahn and Fasshauer, 2012*). Experiments combining $Ca^{2+}$ uncaging and $Ca_i$ measurement in presynaptic terminals with postsynaptic EPSC recording have demonstrated that these sites have a low affinity for $Ca^{2+}$ and fast kinetics (*Bollmann et al., 2000*; *Schneggenburger and Neher, 2000*). The corresponding $Ca^{2+}$-dependent reactions generate a characteristic curve describing the rate of SV exocytosis as a function of $Ca_i$. At low $Ca_i$ values, the response follows a sigmoid rise with a Hill coefficient close to 4. At high $Ca_i$ values, the response saturates, presumably because all available SVs undergo exocytosis. Based on these findings, various multistep kinetic models of SV release have been proposed, where the probability of exocytosis reaches a maximum of 1 for high $Ca_i$ values (*Bollmann et al., 2000*; *Schneggenburger and Neher, 2000*; *Lou et al., 2005*).

The docking/priming steps preceding exocytosis have been extensively studied in recent years. TIRF results using single SV tracking in calyx of Held terminals indicate a sequence of a long-lived (about 3 s long) tethered state (at a distance of <100 nm from the plasma membrane), followed by

*For correspondence:
alain.marty@parisdescartes.fr

Competing interests: The authors declare that no competing interests exist.

shorter-lived (about 0.3 s long) docked, and docked/primed states, before exocytosis (*Midorikawa and Sakaba, 2015*). These results indicate that certain docking/priming steps are much slower than exocytosis. However other results using electrophysiological recordings indicate that in some preparations, the last preparatory step before exocytosis can occur within a few ms only (*Saviane and Silver, 2006*; *Hallermann et al., 2010*; *Miki et al., 2016*; *Kawaguchi and Sakaba, 2017*; *Miki et al., 2018*). In agreement with these studies, a recent electron microscopy study of synaptotagmin-1 modified mouse mutants demonstrates that in these mouse lines, docking of synaptic vesicles occurs within 10 ms following action potential stimulation (*Chang et al., 2018*). Another electron microscopy study carried out on wild type animals showed that following a presynaptic action potential, complex undocking/docking sequences occur on a time scale ranging from 5 to 100 ms (*Kusick et al., 2018*). These results indicate that the last step before exocytosis involves a fast, $Ca^{2+}$-dependent SV movement towards the plasma membrane, and they suggest some degree of integration between preparatory steps and exocytosis (*Neher and Brose, 2018*).

Recently our group has developed methods to investigate SV release in 'simple synapses', small central synapses that contain a single presynaptic active zone (review: *Pulido and Marty, 2017*). Simple synapse recording in cerebellar slices has provided precise information on SV release and has led to several observations indicating that docking sites are not fully occupied at rest, both in GABAergic connections between molecular layer interneurons (MLIs), and in glutamatergic connections between parallel fibers (PFs) and MLIs: (i) in MLI-MLI simple synapses, responses to repeated saturating $Ca^{2+}$ uncaging stimulations display trial to trial fluctuations (*Trigo et al., 2012*); (ii) when testing the responses of simple MLI-MLI synapses to presynaptic action potentials, individual docking sites fluctuate between periods of low release probability and periods of high release probability (*Pulido et al., 2015*); (iii) synaptic facilitation at simple PF-MLI synapses is most simply explained by assuming partial resting docking site occupancy (*Miki et al., 2016*). In spite of these various lines of evidence, however, present estimates for the probability of docking site occupation remain tentative, because of the difficulty in separating this probability from the release probablilty of occupied sites.

Model simulations indicate that the docking site occupancy at rest strongly influences short-term synaptic plasticity and statistics of SV release (*Pulido and Marty, 2018*; *Miki, 2019*). In addition, the proposal of partial docking site occupancy at rest has important implications concerning the maximum response of a synapse. Because a release site needs a docked SV to release, partially occupied docking sites should display a release probability following AP stimulation that remains below 1, even if experimental conditions are chosen to maximize release of docked SVs. To test this prediction, we explore in the present work how high release conditions alter the behavior of simple synapses. We estimate release site numbers in each recording, and obtain absolute numbers for the release probability per release site. We then produce dose-response curves by combining these results with measurements of calcium entry in single varicosities. Comparing the dose-response curve with the predictions of current models of $Ca^{2+}$-dependent exocytosis confirms the notion that docking site occupancy is incomplete at rest, and that partial site occupancy limits the synapse output at high release probability. Furthermore, our results suggest that docking site occupancy grows as a function of the external $Ca^{2+}$ concentration, $[Ca^{2+}]_o$. They further suggest that a given $[Ca^{2+}]_o$, the docking site occupancy can be obtained as the maximal attainable release probability per release site. Finally, our results suggest that previous apparently discrepant results indicating a very high (close to 1) maximal release probability per release site can be explained partially by uncorrected errors in variance-mean analysis, and partially by changes of docking site occupancy with $[Ca^{2+}]_o$.

## Results

### Release probability per docking site during AP trains

In this work we examine counts of SV release events measured in simple PF-MLI synapses when increasing the release probability with several methods. Our aim is to identify the source(s) of the increase of release probability in each case, taking advantage of the detailed information provided by SV counting in individual AZs. As further discussed below, we assume for the interpretation of our results that release occurs at a finite set of docking/release sites, and that SVs undergo

exocytosis only once they are docked. Within this simple docking/release model, we attempt to distinguish increased probability of docking site occupancy by SVs from increased probability of exocytosis of an occupied docking site (*Quastel, 1997*).

*Figure 1A* illustrates the experimental protocol. Having established a recording in an MLI, a pipette was placed either in the granule cell layer or in the molecular layer to stimulate extracellularly a presynaptic granule cell or its associated PF axon. The pipette position and stimulation strength were carefully adjusted to restrict the stimulation to a single presynaptic PF (see Materials and methods; *Malagon et al., 2016*; *Miki et al., 2017*). Trains of 8 action potentials (APs) were applied at a frequency of 100 Hz inside a train, leaving 10 s intervals between trains. Typical traces in response to an 8-AP stimulation train (stimulation times: vertical red bars) are shown in *Figure 1A*. As illustrated in this example, responses to individual APs varied between single quantal responses (with unitary amplitudes on the order of 100 pA or more in this preparation: *Llano and Gerschenfeld, 1993*; examples in responses to 1 st and 2nd AP in 1 st trace), failures (as following 3rd and 4th AP of 1 st trace), and multivesicular responses (eg, following the 5th stimulation in the 1 st trace). Only results passing previously defined criteria for simple synapse recordings were accepted in this work (*Malagon et al., 2016*). Briefly, the criteria were: (i) stability of the recording as a function of time; (ii) amplitude occlusion for consecutive events at short intervals; (iii) low scatter of quantal current amplitudes (*Malagon et al., 2016*). In recordings that met these criteria, we identified single release events during AP trains by deconvolution, using a synapse-specific averaged quantal EPSC profile as template (*Malagon et al., 2016*). We then summed the numbers of SVs released in a 5 ms long time window following each AP stimulation. These numbers represent the output of individual AZs following AP stimulations. They are depicted in matrices as a function of AP number and train number in *Figure 1B*, where grey levels represent various SV numbers.

A first set of experiments was performed under normal release conditions, using an external $Ca^{2+}$ concentration ($[Ca^{2+}]_o$) of 1.5 mM. We then observed a gradual increase in SV counts during a train, reflecting facilitation (*Figure 1B*, upper panel; group data in *Figure 1D*, pink). After obtaining a series of train responses, SV counts were calculated separately for each of the 8 successive AP stimulations, and we calculated means and variances of SV counts according to the AP stimulation number, i. This gave a variance vs. mean plot with 8 data points that could be fitted using a parabola (*Figure 1C*). To interpret these data we assumed that each simple synapse obeys a binomial release process, with N functional units (docking/release sites) acting in parallel (*Malagon et al., 2016*). The intersect of the parabola with the abscissa axis yielded the value of N. For stimulus number i, we calculated the probability $P_i$ that a docking site releases a SV by dividing the mean SV count by N, with a mean N value of 3.6 across synapses. The mean of $P_1$ obtained in this manner under control conditions was 0.089 ± 0.015 while the mean of the maximum of $P_i$ for i = 1–8 was $P_{max}$ = 0.270 ± 0.027. These data show that facilitation markedly increases P under control conditions, with a ratio between $P_1$ and $P_{max}$ of 3-fold. They also indicate that P is far from reaching the maximum value of 1 under these conditions.

## Effect of adding 1 mM TEA on P

Standard docking/release site models assume that a docking site can release only if it is occupied by a SV. Accordingly, these models predict that $P_i$ is the product of $\delta_i$ and $p_i$, where $\delta_i$ is the mean docking site occupancy (sometimes noted $p_{occ}$) before stimulus number i, and $p_i$ is the probability of release of a docked SV (sometimes noted $p_{ves}$) following stimulus number i (*Vere-Jones, 1966*; *Zucker, 1973*; *Quastel, 1997*; *Scheuss and Neher, 2001*). In the following we will accept the $P_i = \delta_i p_i$ relation as a guideline. Recent results calling for a qualification of this hypothesis (*Miki et al., 2018*) will be discussed below.

Increasing evidence indicates that both facilitation and depression mostly follow changes in docking site occupancy, reflecting a trade-off between SV loss by exocytosis and SV gain by docking site replenishment (*Pulido et al., 2015*; *Miki et al., 2016*; *Miki et al., 2018*). Accordingly, we assume in the following that $p_i$ takes a common value, p, independently of i, so that $P_i = \delta_i p$. For i = 1 we write $\delta_1 = \delta$, and $P_1 = \delta p$, where $\delta$ is the resting docking site occupancy. In line with our previous work (*Miki et al., 2016*; *Miki et al., 2017*), we further assume that all release sites have the same release probability and that they are located at equal distances from presynaptic voltage-dependent calcium channels.

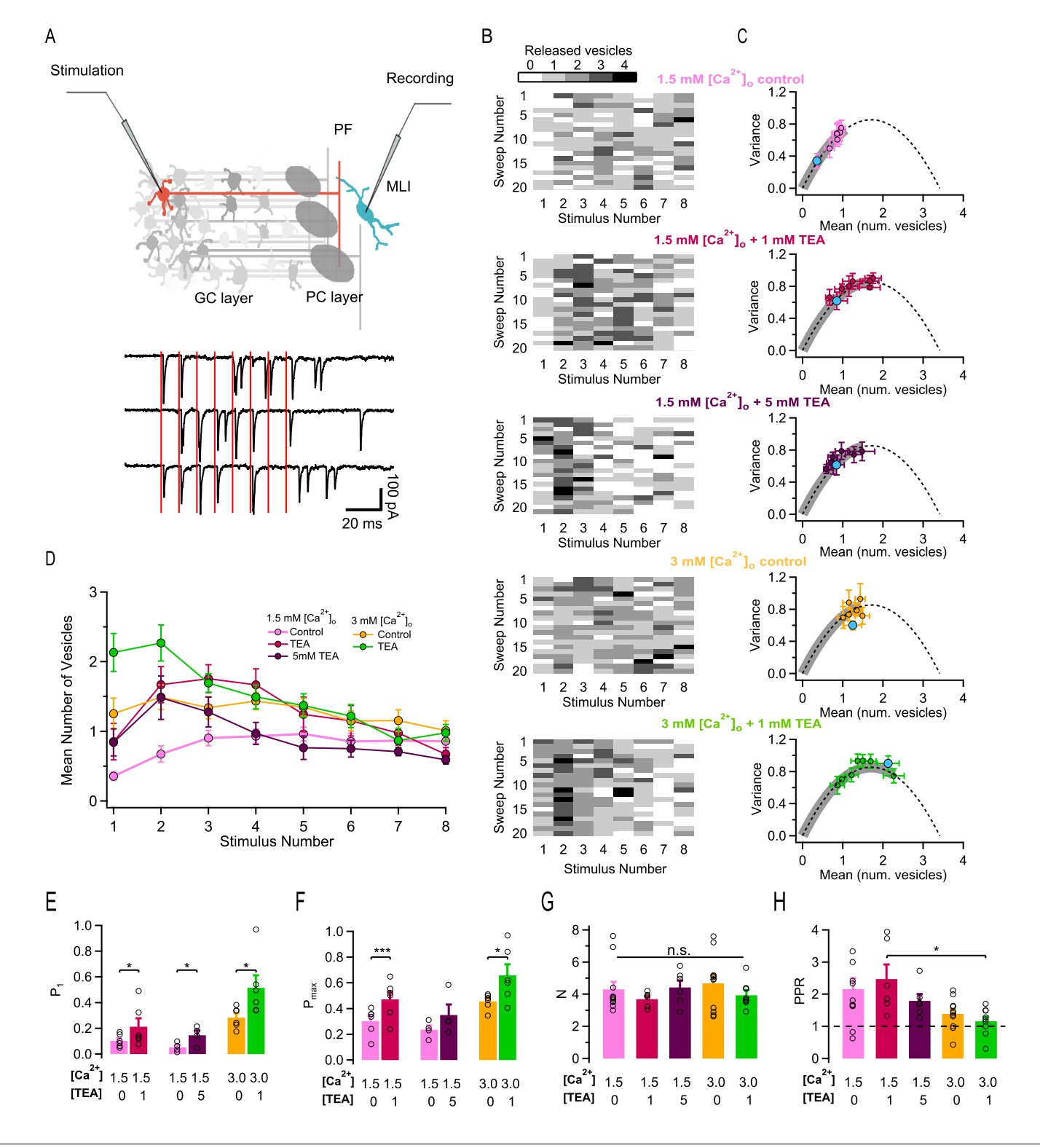

**Figure 1.** Effects of different release conditions on synaptic parameters and short-term plasticity. (**A**) Upper panel: Schematics of recording configuration. A stimulation pipette is positioned close to the soma of a potential presynaptic granule cell while whole-cell recording from a MLI. Careful presynaptic pipette positioning and stimulation strength adjustment results in selective stimulation of a single PF-MLI connection. Lower panel: Individual traces in response to AP trains (8 stimuli, 100 Hz; stimulation timing in red). (**B**) Deconvolution-based counts of released SVs per stimulus. Matrices of SV numbers as a function of AP number for 5 individual experiments are illustrated in different external solutions (from top to bottom: 1.5

*Figure 1 continued on next page*

*Figure 1 continued*

mM $[Ca^{2+}]_o$ in control, in 1 mM TEA, and in 5 mM TEA; 3 mM $[Ca^{2+}]_o$ in control and in 1 mM TEA). Each experiment involved a control period and another set of data in one TEA concentration. Accordingly, the two SV count matrices in 1.5 mM $[Ca^{2+}]_o$ without TEA and with 1 mM TEA stem from the same experiment. Likewise, the two SV count matrices in 3 mM $[Ca^{2+}]_o$ without TEA and with 1 mM TEA stem from the same experiment. (C) Group results of the variance-mean relationship of the number of released vesicles calculated per stimulus (mean ± SEM; results for the first stimulus in blue). The dashed lines represent the parabolic fit after pooling the results in all conditions; it indicates an N = 3.6. Gray thick traces show the excursion over this parabola from the origin until the value that displays the highest P. (D) Time course of SV numbers during a train showing changes in short term synaptic plasticity with $[Ca^{2+}]_o$ and TEA (same data set as in C). (E–H) Group analysis (bars: mean ± SEM, n = 4–12; circles: individual experiments) show an increase of the release probability for the $1^{st}$ stimulus as a function of TEA (E), an increase of the maximal release probability as a function of TEA (F), no change in N (G), and a decrease of the PPR as a function of $[Ca^{2+}]_o$ and TEA (H). Statistical comparisons use paired t-tests in E-F, and unpaired t-tests for all combinations in G-H. No symbol or n.s. = $p > 0.05$; *=$p < 0.05$; **=$p < 0.01$; ***=$p < 0.001$.

The online version of this article includes the following figure supplement(s) for figure 1:

**Figure supplement 1.** Effect of adding TEA on granule cell excitability.

While the value of $P_1$ is readily accessible by performing variance-mean analysis of SV counts, as explained above, separating p from δ poses a challenge. An attractive way to determine δ is to increase the amount of $Ca^{2+}$ entry until p reaches one because in this case, the relation $P_1 = δ p$ becomes $P_1 = δ$. Therefore, measuring $P_1$ under very high release probability conditions provides a direct estimate of δ.

We therefore explored ways to increase p in the present work. Previous studies indicate that addition of TEA increases p in many synapses including PF-MLI synapses by broadening AP duration and increasing the amount of $Ca^{2+}$ entry per AP, mainly by blocking Kv3 channels (*Sabatini and Regehr, 1997*; *Ishikawa et al., 2003*). When applying TEA in granule cell recordings, we found in agreement with previous work (*Sabatini and Regehr, 1997*) a decrease in voltage-dependent $K^+$ current amplitudes and an increase in AP duration (*Figure 1—figure supplement 1*). Furthermore, TEA increased the responses of PF-MLI synapses to AP stimulations (*Figure 1B*: compare first and second panel from above). Comparing released SV counts before and after the addition of TEA in the same experiment, as illustrated in the example of *Figure 1B*, shows a marked increase in the presence of TEA. Average SV counts across experiments for the first AP were $n_1 = 0.36 ± 0.06$ in control and $n_1 = 0.85 ± 0.26$ in TEA (n = 6; p<0.05, paired t-test). Variance-mean analysis of SV counts (*Figure 1C*, first and second graphs from top) showed no change of N upon application of TEA (individual experiments and corresponding means in *Figure 1G*). Given that N stays constant, the change in synaptic strength associated with TEA application is entirely mediated by an increase in P. After normalizing SV counts per docking site, we found that $P_1$ values increased from 0.103 to 0.213 in 1 mM TEA (*Figure 1E*).

Despite this marked increase in $P_1$, the responses to an 8-AP train displayed roughly the same amount of facilitation with and without TEA. Paired-pulse ratio (PPR) values, obtained by calculating the ratio of $n_2/n_1$, were similar with and without TEA (two first columns in *Figure 1H*). When examining the time course of $n_i$ values during the train, the maximum $P_{max}$ was attained earlier in TEA (for i = 3) than in control (for i = 5), but the shapes of the two curves were not strikingly different (compare pink and red curves in *Figure 1D*). These observations are in line with the fact that the value of $P_1$ obtained in TEA (0.213) is far from the maximum attainable value of 1, leaving room for a significant increase of P linked to facilitation.

Finally, the value of $P_{max}$ in the presence of 1 mM TEA was 0.472, far below the maximum of 1. Therefore, increasing release probability by a combination of synaptic facilitation and of TEA application is not sufficient to bring the release probability to 1.

## Saturation of P as a function of TEA concentration

A possible interpretation of the results above is that addition of 1 mM TEA was insufficient to bring the release probability of a docked SV (p) to its maximum of 1. To investigate this possibility, we tested 5 mM TEA in a new series of experiments, starting again from the same 1.5 mM $[Ca^{2+}]_o$ control condition (*Figure 1B*, third row). As documented below, $Ca^{2+}$ entry was substantially increased in 5 mM TEA compared to 1 mM TEA. Nevertheless, even with 5 mM TEA, failures were often observed in response to the 1 st AP (*Figure 1B*, third row), suggesting that $P_1$, the probability that a site releases a SV after the $1^{st}$ AP, was still far from reaching its maximum of 1. In these experiments,

$P_1$ grew from $0.053 \pm 0.018$ without TEA to $0.146 \pm 0.036$ in 5 mM TEA. This represented a mean $P_1$ ratio of 3.0 in individual experiments, not significantly different from the 1.9-fold ratio obtained in the previous experiments with 1 mM TEA (unpaired t-test). Globally these results suggest that TEA increases $P_1$ only up to a certain point.

If p is close to 1 in both 1 mM and 5 mM TEA, then in view of the relation $P_1 = p\, \delta$, $P_1$ is close to $\delta$ in both cases. Therefore, an obvious possible explanation for the limitation of $P_1$ near 0.2 is that this limit represents $\delta$, the docking site occupancy. Starting with a common submaximal $\delta$ value leaves the same room for $\delta$-driven facilitation in 1 mM TEA experiments and in 5 mM TEA experiments. In line with this suggestion, we found similar values for the PPR ($2.47 \pm 0.45$ in 1 mM TEA vs. $1.80 \pm 0.21$ in 5 mM TEA; n. s.) and for $P_{max}$ ($0.472 \pm 0.059$ in 1 mM TEA vs. $0.351 \pm 0.080$ in 5 mM TEA; n. s.) in 1 mM TEA and in 5 mM TEA.

In summary, values of $P_1$, $P_{max}$ and PPR are similar in 1 mM TEA and in 5 mM TEA, suggesting saturation of p as a function of TEA concentration. In 5 mM TEA as in 1 mM TEA, the variance-mean curve remains restricted to the ascending limb of the parabola, where the release probability is less than 0.5 (grey thick lines in *Figure 1C*, 2nd and 3rd panels from above).

## Release statistics as a function of extracellular $Ca^{2+}$ concentration

In view of our failure to reach P = 1 by addition of TEA, we next studied SV release in elevated (3 mM) $[Ca^{2+}]_o$ conditions, both with and without TEA, and we compared the results with those previously obtained under control $[Ca^{2+}]_o$ (1.5 mM).

We first examined results in 3 mM $[Ca^{2+}]_o$ without TEA. In 3 mM $[Ca^{2+}]_o$, $P_1$ was increased compared to 1.5 mM $[Ca^{2+}]_o$ ($0.28 \pm 0.03$ vs. $0.089 \pm 0.015$; $p < 0.01$, unpaired t-test), and it reached a mean value similar to that obtained in 1.5 mM $[Ca^{2+}]_o$ and TEA ($0.28 \pm 0.03$ vs. $0.21 \pm 0.06$). Likewise $P_{max}$ was larger in 3 mM $[Ca^{2+}]_o$ compared to 1.5 mM $[Ca^{2+}]_o$ ($0.46 \pm 0.04$ vs. $0.27 \pm 0.03$; $p < 0.01$, unpaired t-test), and was similar in 3 mM $[Ca^{2+}]_o$ compared to 1.5 mM $[Ca^{2+}]_o$ and TEA ($0.46 \pm 0.04$ vs. $0.47 \pm 0.06$).

The finding that $P_1$ and $P_{max}$ take similar values in 3 mM $[Ca^{2+}]_o$ and in 1.5 mM $[Ca^{2+}]_o$ plus TEA could reflect a common limit to the strength of the synapse. Alternatively, it could be a sheer numerical coincidence. To distinguish between these two possibilities, we next tested the effects of adding 1 mM TEA starting in 3 mM $[Ca^{2+}]_o$. $P_1$ significantly increased from $0.28 \pm 0.03$ to $0.51 \pm 0.09$ ($p < 0.05$, paired t-test; exemplar experiment in lowest 2 panels of *Figure 1B*; group results in *Figure 1E*). Likewise, $P_{max}$ increased from $0.46 \pm 0.04$ to $0.66 \pm 0.08$ ($p < 0.05$, paired t-test; *Figure 1F*). Therefore, whereas neither $P_1$ nor $P_{max}$ increased significantly when increasing the TEA concentration from 1 to 5 mM in 1.5 mM $[Ca^{2+}]_o$ experiments, both $P_1$ and $P_{max}$ were significantly higher in 3 mM $[Ca^{2+}]_o$ and 1 mM TEA compared to 3 mM $[Ca^{2+}]_o$. These results indicate that combining $[Ca^{2+}]_o$ elevation with TEA application is an effective way to overcome the limitations in P apparent at high TEA concentrations with normal $[Ca^{2+}]_o$.

## TEA and $[Ca^{2+}]_o$ increase P by two different mechanisms

In view of the relation $P = \delta\, p$, we propose as an interpretation of the above results that TEA application leads to an increase of p without a change in $\delta$, whereas an increase in $[Ca^{2+}]_o$ results in increases in both $\delta$ and p. In this interpretation, $\delta$ remains constant if $[Ca^{2+}]_o$ is constant, so that even if TEA increases p near its maximum p = 1, P remains limited by $\delta$. This explains the limitation of $P_1$ values near 0.2 in 1.5 mM $[Ca^{2+}]_o$ (*Figure 1E*). By contrast if $[Ca^{2+}]_o$ is increased the value of $\delta$ increases so that the maximum attainable for $P_1$ increases according to the change in $\delta$. This explains the higher $P_1$ values observed when combining TEA and high $[Ca^{2+}]_o$, compared to applying TEA in normal $[Ca^{2+}]_o$ (*Figure 1E–F*).

The proposal of differential effects of TEA and $[Ca^{2+}]_o$ on p and $\delta$ also helps explaining short-term plasticity results shown in *Figure 1*. As detailed earlier (*Pulido and Marty, 2018*), models of synaptic function based on changes in docking site occupancy predict different effects of increasing p or $\delta$ on short-term synaptic plasticity. While all synapses become depressing if $\delta$ increases to the maximum $\delta = 1$, some synapses with low $\delta$ values remain facilitating even if p increases to p = 1. This is because if the initial value of docking site occupancy is low, even if the first AP releases all docked SVs, replenishment of initially empty docking sites can overcompensate the loss of exocytosed SVs, resulting in facilitation. In the present case, while facilitation was prominent in 1.5 mM

$[Ca^{2+}]_o$, without and with TEA, it became weak or absent in 3 mM $[Ca^{2+}]_o$ (yellow and green curves in *Figure 1D*). Simultaneously depression increased, particularly in the presence of TEA (green curve in *Figure 1D*), and the value of the PPR decreased (*Figure 1H*). Overall, the finding that facilitation remains when applying TEA alone is consistent with a pure effect on p, while the more complex effects observed on short-term synaptic plasticity when increasing $[Ca^{2+}]_o$ are consistent with a mixed effect on p and δ.

## Miniature current frequencies in elevated $[Ca^{2+}]_o$ and in TEA

In several synapses including MLI synapses onto Purkinje cells, increasing $[Ca^{2+}]_o$ leads to an increase of miniature current frequency, presumably following an increase in the presynaptic resting $[Ca^{2+}]_i$ (*Llano et al., 2000*). At PF-MLI synapses, we found that miniature EPSC frequency rose from $0.142 \pm 0.033$ Hz in 1.5 mM $[Ca^{2+}]_o$ to $0.364 \pm 0.040$ Hz in 3 mM $[Ca^{2+}]_o$ (p<0.01; n = 6), while mean peak miniature EPSC amplitudes did not change significantly ($93.6 \pm 9.1$ pA in 1.5 mM $[Ca^{2+}]_o$ vs. $105.3 \pm 20.4$ pA in 3 mM $[Ca^{2+}]_o$; p>0.05; n = 6). These data suggest that elevating $[Ca^{2+}]_o$ results in an elevation of presynaptic $[Ca^{2+}]_i$ at PF-MLI synapses. By contrast, we found that TEA (1 mM) did not change miniature EPSC frequency (control frequency: $0.123 \pm 0.014$ Hz; TEA frequency: $0.116 \pm 0.023$ Hz, p>0.05, n = 7). These results suggest that $[Ca^{2+}]_o$ elevation, but not TEA addition, leads to an increase in the presynaptic basal $[Ca^{2+}]_i$. As SV replenishment is enhanced by elevations of $[Ca^{2+}]_i$ (*Neher and Sakaba, 2008*), it is plausible that presynaptic $[Ca^{2+}]_i$ elevation increases the docking site occupancy δ.

## Comparison of variance-mean analyses performed on SV counts and on EPSC peak amplitudes

A striking outcome of the experiments described so far is that in all cases, variance-mean parabolas were far from complete, as illustrated by the thick grey curves in *Figure 1C*. Specifically, in spite of our efforts, we were unable to attain a value of P larger than 0.66 in the above experiments (i. e., the value of $P_{max}$ in 3 mM $[Ca^{2+}]_o$ and 1 mM TEA). This contrasts with many studies reporting P values of 0.9 or larger when performing mean-variance analysis of peak EPSC amplitudes (*Branco and Staras, 2009*). One possible reason for the discrepancy is that P is limited by δ, as mentioned above, and that the value of δ is particularly low in PF-MLI synapses. Another possible reason is methodological. In the traditional method of peak EPSC mean-variance analysis, receptor saturation or desensitization, as well as synaptic jitter, can lead to substantial errors (*Clements, 2003*; *Silver, 2003*). These errors worsen at high release probability, and if they are not corrected, their effects tend to overestimate P and to underestimate N. While methods exist to correct for some of these errors (*Silver, 2003*; *Taschenberger et al., 2005*), corrections are difficult to implement, often requiring dedicated experiments, and they are at best partial. To examine the potential consequences of the errors involved, we next compared in the same recordings the P value obtained from peak EPSC analysis with the value obtained from SV counts.

A comparison between the two kinds of variance-mean analyses is presented in *Figure 2A* in an experiment using 3 mM $[Ca^{2+}]_o$. The variance-mean parabola indicated a $P_1$ value of 0.307 with the SV count method (left, red triangles) and of 0.487 with the peak EPSC method (center, open purple rectangles); $P_{max}$ values were 0.436 with the SV count method and 0.710 with the peak EPSC method. As illustrated in this example, P estimates were invariably higher with the EPSC amplitude analysis than with the SV count analysis. The discrepancy became larger when correcting the variance for EPSC amplitudes for the variations in q values in the same experiment (filled purple rectangles, right panel: $P_1 = 0.530$, and $P_{max} = 0.779$), as is usually done in peak amplitude variance-mean analysis (*Silver, 2003*). To perform the correction, we assumed a CV value of 0.30 for quantal amplitude variance within one synapse, based on previous estimates (*Malagon et al., 2016*). Average results revealed significant discrepancies in $P_1$ estimates in 3 mM $[Ca^{2+}]_o$ ($0.294 \pm 0.032$ with the SV count method vs. $0.612 \pm 0.058$ with the peak EPSC method, p<0.001; *Figure 2B*; note that the present values obtained with the peak EPSC method are in agreement with previous estimates in the same preparation using the same method: *Ishiyama et al., 2014*). Similarly, $P_{max}$ estimates were smaller with the SV count method than with the peak EPSC method ($0.456 \pm 0.025$ vs. $0.806 \pm 0.058$, p<0.01; *Figure 2C*). Estimated N values were larger with the SV counting method than with the peak EPSC method ($3.94 \pm 0.46$ vs. $2.19 \pm 0.29$, p<0.05; *Figure 2D*), suggesting that the peak EPSC

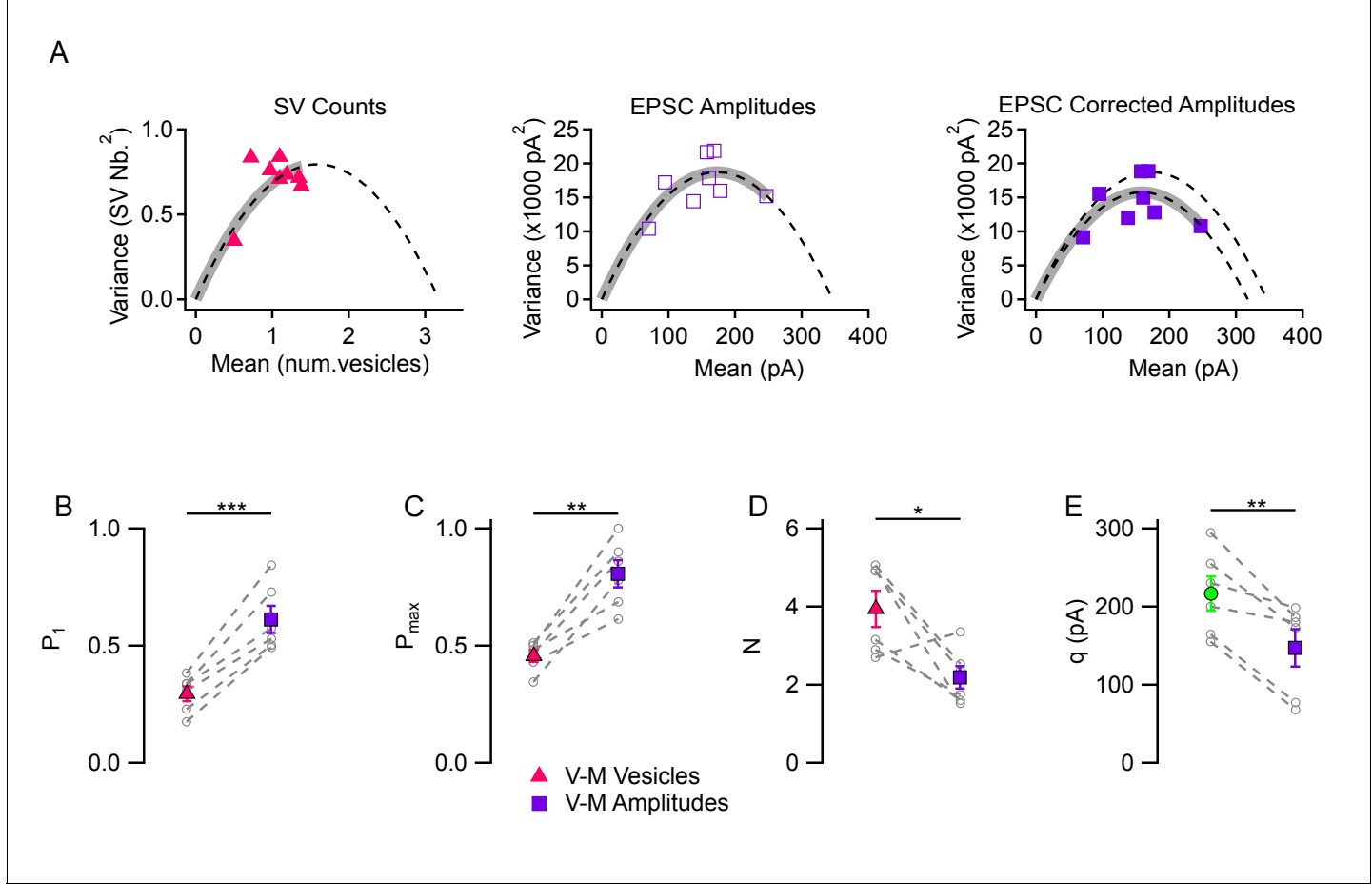

**Figure 2.** Comparison between variance-mean analysis of SV counts and variance-mean analysis of peak EPSC amplitudes. (A) Variance vs. mean plots calculated for different stimulus numbers (8 stimuli per plot) in a representative simple synapse experiment (3 mM [$Ca^{2+}$]$_o$). Using the same data, mean and variance estimates were obtained either using SV counts (left, triangles) or using peak EPSC amplitudes (center and right, squares). Fitted functions follow the equations $f(x) = x - x^2/N$ for SV counts, and $f(x) = x\,q - x^2/N$ for EPSC amplitudes (dashed lines: parabolic fits; gray thick traces show the excursion over the parabola from the origin up to the experimental point with highest P value). The right panel shows the effect of correcting the variance in the peak EPSC analysis for variations in q size ('corrected amplitudes': filled squares). The thick grey parabolas indicate a larger maximum P value with EPSC amplitude analysis than with SV counts, particularly after correction. (B–D) Group analysis of synaptic parameters extracted from 6 experiments as illustrated in (A) using variance-mean analysis based either on SV counts (triangles) or EPSC amplitudes (filled squares; data corrected for q variance). These results show higher estimates of $P_1$ and $P_{max}$, as well as lower N estimates, when using EPSC amplitudes compared to using SV counts. (E) Quantal size estimates using variance-mean analysis of EPSC amplitudes (mean: purple square) are lower than direct measurements obtained for each synapse during delayed release (mean: green circle). In (B–E), error bars show ± SEM, and results from same experiments are linked together with dashed lines. Statistical comparisons show results of paired t-tests between indicated data groups: *=$p < 0.05$; **=$p < 0.01$; ***=$p < 0.001$.

method leads to an underestimate of N. Finally, direct measurements of quantal current amplitudes (217 ± 22 pA) were significantly different from q estimates derived from EPSC amplitude variance-mean analysis (147 ± 24 pA, p<0.01; green vs. purple means in *Figure 2E*). These numbers indicate a significant (about 1.5-fold) under-estimate of q values with the EPSC amplitude variance-mean analysis compared to direct measurements. Altogether these results indicate that the peak EPSC amplitude mean-variance analysis leads to over-estimates of $P_1$ and $P_{max}$, and to under-estimates of N and q, likely due to a combination of receptor saturation and synaptic jitter. Consequently, the difference between the moderate maximum P values found in this work, and higher values found in earlier studies, may partially arise from uncorrected errors linked to receptor saturation and synaptic jitter in these earlier studies.

## Combining measurements of Ca$^{2+}$ entry with P measurements

The above results show that values of P per release site are markedly smaller than 1. If the value of δ were 1, as is commonly assumed, then the low range of P values would predict that P is highly sensitive to Ca$^{2+}$ entry, because the dose-response curve of release probability as a function of [Ca$^{2+}$]$_i$ is cooperative at the foot of the curve (*Dodge and Rahamimoff, 1967*; *Schneggenburger and Neher, 2000*). Therefore we next investigated the changes in Ca$^{2+}$ entry associated with [Ca$^{2+}$]$_o$ changes and with TEA applications. For this purpose, we performed two-photon imaging of single PF varicosities (*Figure 3A*) to determine Ca$^{2+}$ entry elicited by AP trains. We started fluorescence measurements about 30 min after establishing somatic whole-cell recording to allow dye equilibration in the imaged axonal area. Ca$^{2+}$ entry was evaluated as the relative increase of the peak fluorescence measurement of OGB-6F (ΔF/F$_0$: *Figure 3B*). *Figure 3B* shows an experiment where we compared ΔF/F$_0$ signals under control conditions and during TEA application, finding a reversible increase. In conformity with previous results (*Sabatini and Regehr, 1997*; *Brenowitz and Regehr, 2007*; *Miki et al., 2016*), we found that the amount of Ca$^{2+}$ entry per AP did not change significantly during an AP train (*Figure 3C*). Three series of experiments were performed: adding 1 mM TEA starting from a 1.5 mM [Ca$^{2+}$]$_o$ solution, as in panel B (n = 5); increasing the TEA concentration from 1 mM to 5

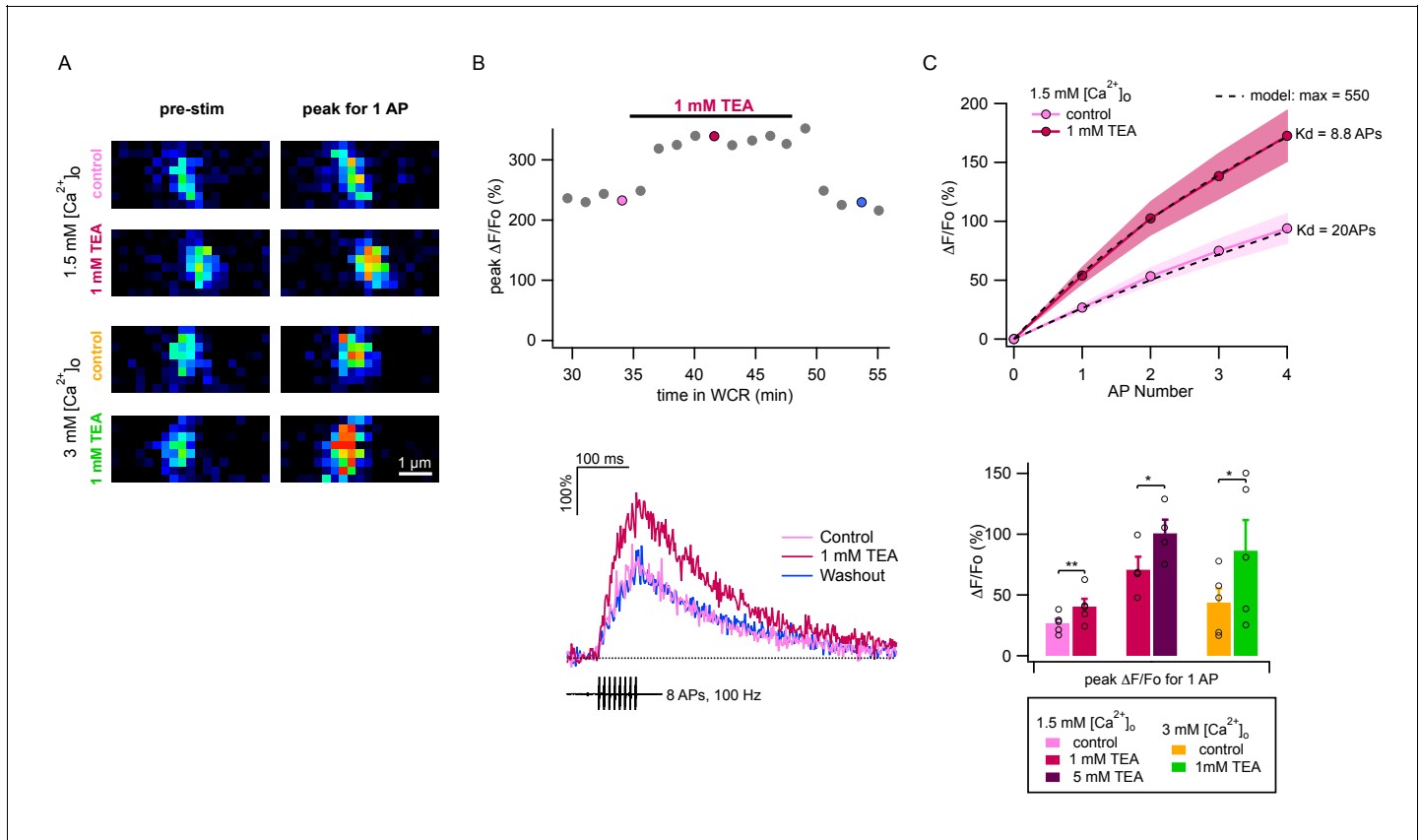

**Figure 3.** TEA augments AP-evoked Ca$^{2+}$ rise in single PF varicosities. (**A**) Single scans of two representative varicosities (one in 1.5 mM [Ca$^{2+}$]$_o$ and the other in 3 mM [Ca$^{2+}$]$_o$), showing [Ca$^{2+}$]$_i$ signal at rest (left) and following a single AP (right), both in control conditions (upper panels) and the presence of 1 mM TEA (lower panels). Cells were loaded with 500 μM of the calcium indicator OGB-6F. In both cases TEA appears to increase the AP-driven [Ca$^{2+}$]$_i$ rise. (**B**) Top: Time-course of peak ΔF/F$_0$ after 8 APs as a function of time in whole-cell recording. After a 30 min loading period the basal fluorescence was stable. 1 mM TEA was added at 35 min, causing an increase in the signal; the effect was reversible upon drug washout. Bottom: Time course of ΔF/F$_0$ signal for individual responses during and after 8 AP stimulation; corresponding dots with same color coding in the upper plot. (**C**) Top: Average AP-evoked peak ΔF/F$_0$ values as a function of AP number (shadowed colors indicate ± SEM; n = 5 varicosities in control, and n = 10 in 1 mM TEA). Dotted lines show fits of the data assuming constant Ca$^{2+}$ entry per AP, using a hyperbolic function describing the sensitivity of OGB-6F on Ca$^{2+}$ concentration (see Materials and methods; half saturation points are indicated next to each trace). Bottom: Average peak ΔF/F$_0$ evoked by the first AP for each treatment. Circles show results from individual experiments. Bars show means ± SEM. Significant differences between mean values are illustrated (*: p<0.05; **: p<0.01; paired t-tests).

mM, starting in a 1.5 mM $[Ca^{2+}]_o$ plus 1 mM TEA solution (n = 4); and adding 1 mM TEA in 3 mM $[Ca^{2+}]_o$ (n = 5). Group results showed significant increases of $Ca^{2+}$ entry upon application of 1 mM TEA, both when starting from a 1.5 mM $[Ca^{2+}]_o$ solution (control: 26.7 ± 3.6%; 1 mM TEA: 40.5 ± 3.6%; p<0.01, paired t-test) and when starting from a 3 mM $[Ca^{2+}]_o$ solution (control: 43.8 ± 11.6%; 1 mM TEA: 86.4 ± 25.2%; p<0.05; paired t-test; *Figure 3C*, lower panel). Likewise, we found a significant increase of fluorescence measurements in 1.5 mM $[Ca^{2+}]_o$ when increasing the TEA concentration from 1 to 5 mM (1 mM TEA: 70.8 ± 10.6%; 5 mM TEA: 100.7 ± 11.3%; p<0.05; paired t-test; *Figure 3C*, lower panel). Therefore, even though our previous results indicate that $P_1$ does not increase when elevating the TEA concentration from 1 to 5 mM, the present imaging results show that $Ca^{2+}$ entry is substantially larger at the higher TEA concentration. The opposite situation is found when comparing results in the presence of 1 mM TEA and 3 mM $[Ca^{2+}]_o$ (green bar) with those in the presence of 5 mM TEA and 1.5 mM $[Ca^{2+}]_o$ (purple bar). Whereas $P_1$ is larger in the first solution than in the second (*Figure 1E*), $\Delta F/F_0$ signals appear similar (*Figure 3C*). Therefore altogether, comparison of the results in *Figure 1E* and *Figure 3C* reveals discrepancies between $P_1$ results and $\Delta F/F_0$ results, indicating that p, the release probability of docked SVs, is not the only determinant of $P_1$.

## Modelling the $P_1([Ca^{2+}]_i)$ relation

Current models of SV release assume that a presynaptic AP induces a local $[Ca^{2+}]_i$ transient at the level of vesicular $Ca^{2+}$ sensors with a sub-ms duration and an amplitude of several 10 s of µM (*Bollmann et al., 2000*; *Schneggenburger and Neher, 2000*). The $Ca^{2+}$ sensors respond to the $[Ca^{2+}]_i$ transient with a sequence of fast $Ca^{2+}$ binding to a series of 4–5 cooperative binding sites, followed by an irreversible exocytosis step (*Dodge and Rahamimoff, 1967*; *Wu and Saggau, 1994*; *Bollmann et al., 2000*; *Schneggenburger and Neher, 2000*; *Lou et al., 2005*). We found that depending on the mode of stimulation, such models predict different dose-response curves representing the synapse output as a function of peak global $[Ca^{2+}]_i$. In *Figure 4A–B*, we compare the dose-response curve for step $[Ca^{2+}]_i$ increases (*Figure 4A*) or for AP stimulations (*Figure 4B*), using $[Ca^{2+}]_{Ii}$-sensitive reaction steps and $[Ca^{2+}]_i$ profiles previously developed to simulate release at PF-MLI synapses (*Miki et al., 2018*). Even though $Ca^{2+}$-sensitive reaction steps are the same for the two sets of simulations both using a modified version of the allosteric model of *Lou et al. (2005)*, as explained in *Miki et al. (2018)*, the dose-response curve is markedly steeper in response to step $[Ca^{2+}]_i$ increases (with an apparent Hill coefficient n = 3; *Figure 4A*) than in response to AP stimulations (with an apparent Hill coefficient n = 1.7; *Figure 4B*). The difference reflects the fact that in AP-induced responses, effective calcium levels vary in a complex manner with calcium entry, due to the diffusion of entering calcium ions and of their interaction with a variety of calcium buffers. These simulations indicate that the Hill coefficient of the dose-response curve does not depend only on the number of $Ca^{2+}$ binding steps in the kinetic model (5 in the present case), but also on the mode of stimulation ($Ca^{2+}$ uncaging vs. AP).

Next, we asked how the dose-response curve of *Figure 4B* could be used to estimate δ. For this purpose, we combined together $P_1$ values obtained under the various experimental conditions of *Figure 1*, together with peak global $[Ca^{2+}]_I$ data obtained from *Figure 3*, creating an experimental plot of $P_1$ as a function of peak global $[Ca^{2+}]_I$ (*Figure 4C*). In this plot, $P_1$ values are overall means derived from the paired control/TEA experiments of *Figure 1* as well as from other experiments including the TEA/(TEA + 4-AP) experiments to be described below. In addition, $P_1$ values were corrected to only incorporate release events coming directly from docking sites ('one-step release'), as opposed to indirect release events coming from replacement sites ('two-step release': *Miki et al., 2018*). Because 2-step release events do not originate from docked SVs, they should not be considered in the estimate of δ. Fortunately, in ordinary experimental conditions, 2-step release appears only starting from the 2nd AP in a train, and is not detectable after the 1st AP. In these cases, no correction for $P_1$ is needed to account for 2-step release. Only in the most extreme condition of 3 mM $[Ca^{2+}]_o$ + TEA was a correction necessary (by 18%). Also, peak global values from *Figure 3C* were corrected for partial dye saturation. To relate the corrected experimental results in *Figure 4C* to the simulation in *Figure 4B*, we note that in *Figure 4B*, the ordinate is p, the release probability of docked SVs, whereas in *Figure 4C*, the ordinate is $P_1$, the global release probability in response to the first stimulus in a train. As already pointed out, $P_1$ data points incorporate both δ and p factors, according to the relation $P_1 = \delta p$. Clearly, $P_1$ data differ from the curve in *Figure 4B* and

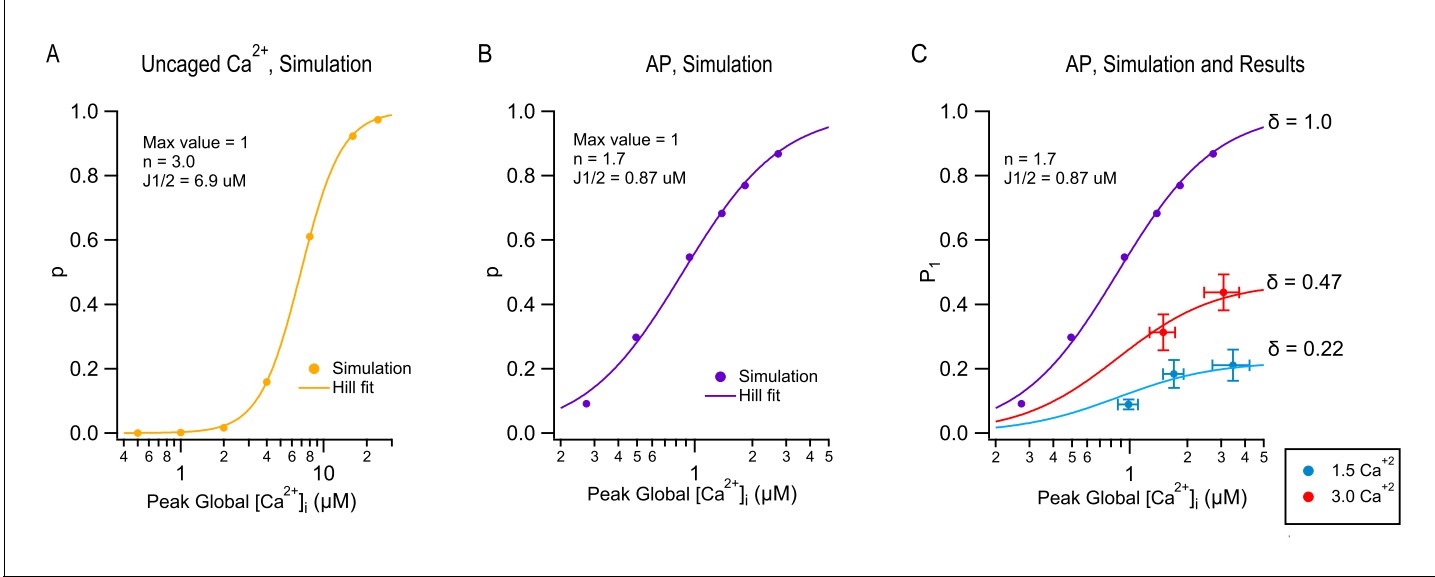

**Figure 4.** Modeling TEA-induced P increases in 1.5 mM $[Ca^{2+}]_o$ and in 3 mM $[Ca^{2+}]_o$. (A) Plot of the release probability (p) as a function of step $[Ca^{2+}]_i$ increases, as predicted by the allosteric vesicular release model of *Lou et al. (2005)*, modified as explained in *Miki et al. (2018)*. (B) The same model predicts a different dose-response curve when plotting the release probability of a docked SV as a function of peak global $[Ca^{2+}]_i$ following an AP stimulation, by varying the total amount of $Ca^{2+}$ entry per AP. For this simulation the $Ca^{2+}$ diffusion and buffering model developed earlier for PF-MLI synapses was used (*Miki et al., 2018*). Note the more shallow dose-response curve (Hill coefficient: 1.7 vs. 3.0) and higher apparent affinity ($J_{1/2}$: 0.87 µM vs. 6.9 µM) with AP stimulation compared to step $[Ca^{2+}]_i$ stimulation. (C) Combining together the results of *Figure 1* and of *Figure 3* for the first stimulation in a train produces a plot of $P_1$ as a function of peak global $[Ca^{2+}]_i$, both at 1.5 mM $[Ca^{2+}]_o$ (blue: no TEA, 1 mM TEA, and 5 mM TEA) and at 3 mM $[Ca^{2+}]_o$ (red: no TEA, and 1 mM TEA). These data cannot be fit directly with the dose-response curve in (B) (purple), but they can be fit using two different scaled versions of this curve, giving $\delta$ = 0.22 in 1.5 mM $[Ca^{2+}]_o$ and $\delta$ = 0.47 in 3 mM $[Ca^{2+}]_o$.

The online version of this article includes the following figure supplement(s) for figure 4:

**Figure supplement 1.** Effect of 4-AP on release.

furthermore, they cannot be fit from this curve by using a single scaling factor. However, acceptable fits are obtained when using two different scalings depending on $[Ca^{2+}]_o$ (*Figure 4C*). In these plots, $\delta$ appears as the asymptotic value of $P_1$ at high global $[Ca^{2+}]_i$, yielding $\delta$ = 0.22 at 1.5 mM $[Ca^{2+}]_o$ and $\delta$ = 0.47 at 3 mM $[Ca^{2+}]_o$ (*Figure 4C*). Altogether this analysis suggests that $\delta$ values increase as a function of $[Ca^{2+}]_o$ and therefore, that changes in both $\delta$ and p contribute to the sensitivity of P as a function of $[Ca^{2+}]_o$.

## Combined effects of TEA and 4-AP

It may be asked whether the above results may be extended to other blockers of potassium channels than TEA. Both in MLI terminals and in the calyx of Held, 4-AP increases synaptic transmission more efficiently than TEA (*Tan and Llano, 1999*; *Ishikawa et al., 2003*). In MLI terminals, 4-aminopyridine (4-AP) markedly increases calcium entry whereas TEA alone is almost ineffective; combining TEA and 4-AP together leads to an even larger AP-induced calcium elevation than when applying 4-AP alone (*Tan and Llano, 1999*). In view of these earlier results, we expected that combining TEA and 4-AP in our preparation should lead to a very large presynaptic calcium entry. Nevertheless, supposing that the model of *Figure 4* is valid, $P_1$ should not be able to increase beyond the value of $\delta$. Group results showed $P_1$ values of 0.140 ± 0.053 in 1.5 mM $[Ca^{2+}]_o$ and 1 mM TEA without 4-AP, and of 0.187 ± 0.048 after further addition of 15 µM 4-AP (*Figure 4—figure supplement 1*). After adding 4-AP, both 2-step release and facilitation were very pronounced, probably due to a marked increase in calcium entry. After correction for 2-step release, $P_1$ in TEA + 4-AP was 0.125 ± 0.032, a value similar to that in TEA alone. Meanwhile, $P_{max}$ increased from 0.263 ± 0.046 in TEA alone to 0.550 ± 0.149 in TEA + 4-AP (*Figure 4—figure supplement 1*), a value similar to that obtained in 3 mM $[Ca^{2+}]_o$ and 1 mM TEA (*Figure 1*). Thus, while $P_1$ was limited by the resting docking site occupancy, the synapse could deliver more SVs upon repetitive stimulation as SV replenishment was

increased. In conclusion, whether calcium entry is increased with TEA alone or with a combination of TEA and 4-AP, $P_1$ remains limited by the value of $\delta$.

## Discussion

### Main finding

In the present work we take advantage of the recently developed method of SV counting (*Malagon et al., 2016*) to examine variations of the release probability of single docking sites (P) during short term synaptic plasticity, following addition of TEA, and when changing $[Ca^{2+}]_o$. Our main finding is that the synaptic strength at the beginning of an AP train ($P_1$) has a maximum attainable value that depends on the value of $[Ca^{2+}]_o$. With 1.5 mM $[Ca^{2+}]_o$, the maximum of $P_1$ is 0.22 SV per AP and per release/docking site, whereas with 3 mM $[Ca^{2+}]_o$, the maximum of $P_1$ is 0.47.

Our interpretation of this finding is that when the release probability of docked SVs, p, is close to 1, $P_1$ is limited by the resting docking site probability $\delta$, and that $\delta$ is lower in 1.5 mM $[Ca^{2+}]_o$ compared to 3 mM $[Ca^{2+}]_o$. The maximum output per site gives an estimate of resting docking site occupancy, a basic parameter of synaptic function that has been difficult to evaluate.

### Correcting release probability for docking site occupancy

Our results confirm earlier suggestions that $\delta$ is less than 1 at PF-MLI synapses (*Miki et al., 2016*; *Miki et al., 2018*). In addition, we now provide reliable $\delta$ estimates. The present finding that $\delta = 0.22$ under physiological conditions (1.5 mM $[Ca^{2+}]_o$) implies that there is a large discrepancy between the release probability per docking site, $P_1$, and the release probability per occupied docking site, p. By inverting the $P_1 = \delta p$ relation, we obtain from $P_1 = 0.089$ and $\delta = 0.22$ that $p = 0.39$ in 1.5 mM $[Ca^{2+}]_o$. At 3 mM $[Ca^{2+}]_o$, we have $P_1 = 0.28$ and $\delta = 0.47$, so that $p = 0.60$. Therefore both at 1.5 mM $[Ca^{2+}]_o$ and at 3 mM $[Ca^{2+}]_o$, the actual release probability of docked SVs is substantially higher than it would appear without the correction. In the presence of TEA, p values calculated from the pooled data of *Figure 4* approach 1 (1.5 mM $[Ca^{2+}]_o$: $p = 0.85$ in 1 mM TEA and $p = 0.95$ in 5 mM TEA; 3 mM $[Ca^{2+}]_o$: $p = 0.89$ in 1 mM TEA, value corrected for 2-step release). Therefore in the present preparation, $P_1$ values in the presence of TEA give reasonably good approximations of $\delta$.

This correction is important when relating release probability and local calcium profiles. For example, at synapses between PFs and Purkinje cells, P values are comparatively small, yet EGTA application experiments suggest tight coupling between SVs and $Ca^{2+}$ entry, indicating high p values (*Schmidt et al., 2013*). A low $\delta$ value at these synapses, similar to the value found here for PF-MLI synapses, is a plausible explanation for these apparently discrepant results.

### Comparison with other preparations

Increasing evidence suggests that part of inter-synaptic variability may reside in differences in docking site occupancy. Specifically, synapses follow a gradient from low release probability, facilitating synapses to high release probability, depressing synapses, presumably depending on the proportion of occupancy of release sites by SVs (*Pan and Zucker, 2009*; *Pulido and Marty, 2018*; *Neher and Brose, 2018*). PF-MLI synapses have a low $\delta$ value in normal physiological conditions (0.22 at 1.5 mM $[Ca^{2+}]_o$ according to the present study) and are facilitating, indicating that they belong to the first category, called 'tonic synapses' (*Pan and Zucker, 2009*; *Neher and Brose, 2018*). By contrast, SV release statistics at simple MLI-MLI synapses indicate a higher resting $\delta$ value near 0.5–0.7 in 2 mM $[Ca^{2+}]_o$ (*Trigo et al., 2012*; *Pulido et al., 2015*). Furthermore MLI-MLI synapses are depressing, and altogether MLI-MLI synapses appear closer to the 'phasic synapse' category (*Pan and Zucker, 2009*; *Neher and Brose, 2018*).

Because of the difficulty of separating $\delta$ from p, estimates of $\delta$ in most preparations are not available. Nevertheless, earlier variance/mean analysis studies, often carried out on peak synaptic current amplitudes, give constraints on $\delta$. Maximum P values reported in these studies are usually in the range 0.5–1 (*Branco and Staras, 2009*). In view of the relation $P = \delta p$, since $p<1$, such large P values suggest resting $\delta$ values above 0.5. However these maximum P values are almost always obtained in high $[Ca^{2+}]_o$ conditions (calyx of Held: *Taschenberger et al., 2005*; PF-MLI synapses: *Ishiyama et al., 2014*; fly neuromuscular junction: *Reddy-Alla et al., 2017*; hippocampal cultures:

*Sakamoto et al., 2018*). In view of the present evidence that δ increases as a function of $[Ca^{2+}]_o$, the high P values at high $[Ca^{2+}]_o$ are compatible with δ values < 0.5 under normal $[Ca^{2+}]_o$ conditions, at least in some (tonic) synapses. Apart from using high $[Ca^{2+}]_o$ conditions, another factor that may have contributed in some studies to high apparent P values is uncorrected synaptic jitter and receptor saturation, as documented above (*Figure 2*). One notable exception is the climbing fibre-Purkinje cell synapse, that may represent an extreme case of phasic synapses, and that exhibits P values above 0.9 at normal $[Ca^{2+}]_o$ (*Silver et al., 1998*). Overall our conclusions appear well compatible with earlier P estimates obtained with variance/mean analysis of peak synaptic current amplitudes.

## Possible mechanisms of changes of δ with $[Ca^{2+}]_o$

An increase in $[Ca^{2+}]_o$ likely acts on the docking site occupancy by inducing an increase in $[Ca^{2+}]_i$ after reequilibration of the cytosolic $Ca^{2+}$ concentration. It is widely accepted that following docking site depletion, docking site replenishment is sensitive to the bulk $[Ca^{2+}]_i$ (*Neher and Sakaba, 2008*). This suggests that the proportion of occupied docking sites at rest may be enhanced by a $[Ca^{2+}]_i$ increase, as the equilibrium is shifted in favor of SV binding to the docking site. Recent electron microscopy studies are in accord with this proposal (*Chang et al., 2018*; *Kusick et al., 2018*). Several AZ molecules could be responsible for $Ca^{2+}$-dependent SV docking, including synaptotagmin-1 (*Chang et al., 2018*) and Munc13 (*Shin et al., 2010*). Another possibility is that $Ca^{2+}$-dependent SV docking is due to activation of actomyosin-driven SV movement by $[Ca^{2+}]_i$ (*Lee et al., 2012*; *Miki et al., 2016*).

## Physiological relevance

A $[Ca^{2+}]_i$ –induced change in resting docking site occupancy may not only apply when changing $[Ca^{2+}]_o$, as proposed in the present work, but it may also occur following a number of other manipulations leading to a change in the global presynaptic $[Ca^{2+}]_i$ concentration. In the calyx of Held, as well as in MLI terminals, subthreshold presynaptic depolarization activates low-threshold voltage-gated $Ca^{2+}$ channels, thus increasing release probability (*Awatramani et al., 2005*; *Bouhours et al., 2011*). Activation of Cl⁻-permeant presynaptic glycinergic or GABAergic receptors in these synapses engages this pathway, leading to a presynaptic $[Ca^{2+}]_i$ increase and to an enhancement of synaptic strength (*Turecek and Trussell, 2001*; *Trigo et al., 2007*; *Zorrilla de San Martin et al., 2017*). Activation of cationic channels by presynaptic nicotinic receptors, AMPA receptors, or NMDA receptors, is also able to increase presynaptic $[Ca^{2+}]_i$, in various preparations including MLI terminals, and these actions have been shown to increase release (*Sharma and Vijayaraghavan, 2003*; *Rossi et al., 2008*; *Rossi et al., 2012*). Finally, enhancement of neurotransmitter release by passive spread of subthreshold somatic depolarization in hippocampal and cortical synapses ('analog signaling') depends on an increase in presynaptic $[Ca^{2+}]_i$ concentration (*Shu et al., 2006*; *Alle and Geiger, 2006*). In view of the present results, it is possible that these effects are mediated at least in part by an increase in resting docking site occupancy. Altogether, $[Ca^{2+}]_i$-driven changes of docking site occupancy appears as a simple and powerful mechanism of gain control of synaptic strength that may be used by various forms of neuromodulation.

## Materials and methods

### Recording procedures

Sagittal slices (200 μm thick) were prepared from the cerebellar vermis of Sprague-Dawley rats (P12–P17) following the animal care guidelines of Paris Descartes University (approval no. A-750607). Recordings were obtained from either basket or stellate cells in the molecular layer; these cells were collectively called molecular layer interneurons (MLIs). The composition of the extracellular solution was (in mM): 130 NaCl, 2.5 KCl, 26 NaHCO₃, 1.3 NaH₂PO₄, 10 glucose, 2 CaCl₂, and 1 MgCl₂ (osmolarity: 300 mosm). This solution was equilibrated with 95% O₂ and 5% CO₂ (pH 7.4). The internal recording solution contained (in mM): 144 K-gluconate, 6 KCl, 4.6 MgCl₂, 1 EGTA, 0.1 CaCl₂, 10 HEPES, 4 ATP-Na, 0.4 GTA-Na; pH 7.3 (osmolarity: 300 mosm). Recordings were at 30–34°C.

## Simple synapse recording

To study simple PF-MLI synapses, potential postsynaptic MLIs were recorded under voltage clamp at –60 mV. For extracellular stimulation, a monopolar pipette was filled with internal solution. NMDA receptors and GABA$_A$ receptors were blocked by inclusion of D(–)−2-amino-5-phosphonopentanoic acid (APV, 50 µM) and gabazine (15 µM). Two alternative procedures were used to find an appropriate location for electrical stimulation. In the first procedure, an electrical stimulation pipette was located in the molecular layer, above the dendritic field of the recorded MLI, and the pipette position and stimulation strength were adjusted to obtain minimal stimulation (*Miki et al., 2016*). In the second procedure, we puff-applied the internal solution including 150 mM K$^+$ from a pipette using small pressure steps while moving the pipette in the granule cell layer (*Miki et al., 2017*). When we found a burst-like EPSC response in the postsynaptic cell, we reduced the pressure of puffing to better define the spot for stimulation. Then we switched to electrical stimulation, using the same pipette, and we adjusted stimulation intensity to fire a connected granule cell under minimal stimulation conditions. With either procedure, definitive acceptance of the experiment as a usable simple synapse recording occurred after analysis and depended on three criteria (*Malagon et al., 2016*): (i) a decrement of EPSC amplitudes of second events in a pair, reflecting activation of a common set of receptors belonging to one PSD; (ii) a Gaussian distribution of EPSC amplitudes with a coefficient of variation (CV) less than 0.5; and (iii) stability of the overall responsiveness over time. Single stimulations and trains of two or eight stimulation pulses were applied repetitively with intervals of 10 s between sweeps. Statistical data were derived from sequences of 10–30 trains.

## Decomposition of EPSCs

We determined occurrence times of individual EPSCs based on deconvolution analysis, as detailed in *Malagon et al. (2016)*, and we built latency distributions by averaging the occurrence times across experiments. We briefly describe the analysis here. Firstly, we made an average of single EPSCs obtained during asynchronous release to obtain a template in a given synapse. Then the average mEPSC was fit by triple-exponential function with five free parameters (rise time, amplitude, fast decay time constant, slow decay time constant, and amplitude fraction of slow decay). Next, mEPSC and individual data traces were deconvolved using the five parameters. The deconvolved mEPSC resulted in a narrow spike, and the deconvolved data traces resulted in sequences of spikes. Finally, we fit a given deconvolved trace by a sum of scaled narrow spikes in order to obtain the timing of each event. The amplitude parameter was free because the peak EPSC amplitude varied during a train due to receptor saturation and desensitization. The above procedure had a detection limit that caused a failure of separation of two events occurring within 0.2 ms. To correct for missed events, we split into two the events having amplitudes at least 1.7 times larger than the average amplitude obtained during asynchronous release.

## Pharmacological manipulations

In TEA experiments, we checked somatic potassium currents by applying 3 ms voltage step to 0 mV until TEA reduced potassium current amplitudes to a stable level, and then we started to collect data.

## Ca$^{2+}$ imaging of presynaptic varicosities

For Ca$^{2+}$ imaging experiments, sagittal (200 µm) slices were prepared using a modified extracellular saline, as detailed in *Brenowitz and Regehr (2007)*. Experiments were conducted in the same conditions as in the electrophysiology experiments (32–34°C, with the 3 mM extracellular Ca$^{2+}$ saline including APV and gabazine). Granule cells were loaded under whole-cell recording with a solution containing (in mM): 140 K-gluconate, 5.4 KCl, 4.1 MgCl$_2$, 9.9 HEPES, 0.36 Na-GTP, 3.6 Na-ATP, 500 µM of Oregon green 488 BAPTA-6F (OGB-6F: K$_d$ for calcium of 5.1 µM; Invitrogen) and 20 µM Alexa-594 (Invitrogen). Imaging was performed with a custom-built 2-photon system, with 820 nm excitation provided by a MaiTai-Sapphire laser (Spectra Physics, USA). In order to visualize the granule cell axon, large raster scans were performed while acquiring the Alexa 594 fluorescence with a red channel photomultiplier (Hamamatsu H7422 PA-sel, bandpass emission filter 635 ± 65 nm, Chroma Technology; or avalanche photodiode Perkin Elmer, SPCM-AQR-13). Single varicosity imaging was performed using raster scans of 5 by 2 µm dimensions at dwell times of 2 ms. The granule

cells were kept under current clamp conditions. We found that keeping a hyperpolarized holding potential improved recording stability, so that resting membrane potential was kept around –90 mV. APs were evoked by 1 ms steps of 350–500 pA. Stimulation protocols were 4 or 8 APs at 100 Hz and were repeated every 1 min. Calcium signaling was analyzed in the pixels encompassing the varicosity in terms of fluorescence changes relative to pre-stimulus values ($\Delta F/F_o$, expressed in %) with software written in the Igor Pro programming environment (Wavemetric, Lake Oswego, OR, USA).

To fit the data in *Figure 3C*, we assumed gradual saturation of OGB-6F as a function of $Ca^{2+}$ entry, with equal contributions for each AP (*Miki et al., 2016*). Estimated $K_d$ values are indicated in *Figure 3C*, upper panel. As before, dye saturation was set at 6.5 times the baseline level (*Miki et al., 2016*). The data in *Figure 4C* are based on the pooled data in the lower panel of *Figure 3C*, after correcting for dye saturation.

## Simulations

We calculated the release probabilities for various $Ca^{2+}$ waveforms by numerical integration of differential equations using Igor Pro with a time interval of 0.001 ms as described (*Miki et al., 2018*). We performed the simulations using a simple one-step release model without replenishment at docking sites (*Miki et al., 2018*), *Figure 1*) with the allosteric model of *Lou et al. (2005)*. According to the fit of data in *Miki et al. (2018)*, we used the following parameter values for the allosteric model: $K_{on} = 5 \times 10^8$ $M^{-1}s^{-1}$, $K_{off} = 5000$ $s^{-1}$, b = 0.75, $\gamma = 1500$ $s^{-1}$, and f = 31.1. $\gamma$ is the fusion rate of the 5Ca-binding state ($V_{5Ca}$), which is identical to $l_+ \times f^5$. f is a factor determining the increase in vesicle fusion rate upon $Ca^{2+}$ binding. The fusion rate at $V_{0Ca-4Ca}$ is $l_+ \times f^{0-4}$. b is a cooperativity factor (*Lou et al., 2005*). In *Figure 4A-B*, the probability of occupancy at docking site was set to 1. For the $Ca^{2+}$ uncaging simulation, we assumed step $[Ca^{2+}]$ increases from a basal $[Ca^{2+}]$ of 50 nM to 0.5, 1, 2, 4, 8, 16, or 24 µM, and we calculated the release probabilities by dividing the total number of vesicle release between 0 and 5 ms by the docking site number. For the AP simulation, we used the local $[Ca^{2+}]$ obtained from $Ca^{2+}$ simulation at 40 nm distance from the nearest $Ca^{2+}$ channels (*Miki et al., 2018*). Global $[Ca^{2+}]$ was obtained at the center of a bouton in $Ca^{2+}$ simulation. The $Ca^{2+}$ simulation was described in detail in a previous report (*Miki et al., 2018*). Briefly, we calculated the spatiotemporal distribution of $[Ca^{2+}]$ at a PF bouton. We used the bouton size of $0.9 \times 0.5 \times 0.5$ µm³ cuboid, and the $Ca^{2+}$ channel distributions based on the electron microscopy observations (*Miki et al., 2017*; *Miki et al., 2018*). The amplitude of $Ca^{2+}$ influx through each channel during an AP was set at 0.2 pA and a fraction of channel open probability during an AP of ~0.7. We assumed a Gaussian-shaped $Ca^{2+}$ influx with a half-amplitude duration of 0.34 ms. Other model parameters were set following *Miki et al. (2018)*. To create $Ca^{2+}$ waveforms having various amplitudes for the AP simulation, we multiplied the simulated local and global $[Ca^{2+}]$ by a factor of 0.25, 0.5, 1, 1.5, 2 and 3 while keeping the resting $[Ca^{2+}]$ of 50 nM. We calculated the release probabilities by dividing the total number of vesicle release between 0 and 5 ms after the beginning of the local $[Ca^{2+}]$ increase by the number of docking sites.

## Acknowledgements

We thank Isabel Llano for her advice and support in two-photon imaging experiments. This work was supported by CNRS (UMR 8118, and UMR 8003), by the European Community (ERC Advanced Grant 'Single Site' to AM, nb. 294509), by JSPS (KAKENHI Grant JP18K06472 to TM, and Core-to-Core Program A, Advanced Research Networks), and by Fondation pour la Recherche Médicale (grant SPF201809007190).

## Additional information

### Funding

| Funder | Grant reference number | Author |
| --- | --- | --- |
| Centre National de la Recherche Scientifique | UMR 8003 | Alain Marty |
| European Research Council | Advanced Grant Single Site 294509 | Alain Marty |

| JSPS | KAKENHI Grant JP18K06472 | Takafumi Miki |
| Fondation pour la Recherche Médicale | SPF201809007190 | Van Tran |

The funders had no role in study design, data collection and interpretation, or the decision to submit the work for publication.

## Author contributions

Gerardo Malagon, Formal analysis, Investigation, Writing; Takafumi Miki, Formal analysis, Investigation, Writing, Funding; Van Tran, Funding acquisition, Investigation, Writing - review and editing; Laura C Gomez, Investigation, Data analysis; Alain Marty, Conceptualization, Project administration, Writing, Funding

## Author ORCIDs

Alain Marty (ID) https://orcid.org/0000-0001-6478-6880

## Decision letter and Author response

Decision letter https://doi.org/10.7554/eLife.52137.sa1
Author response https://doi.org/10.7554/eLife.52137.sa2

## Additional files

### Supplementary files

- Source data 1. Igor figure files.
- Transparent reporting form

### Data availability

An additional file called 'Source data 1' contains original data as well as analysis for this article.

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
