## [Decision Letter]

**Acceptance summary:**

This paper addresses a classical problem in the performance of a synapse. Using a recently published method for counting released synaptic vesicles, the authors have studied variations of release probability of individual docking sites by changing the extracellular calcium concentration while blocking voltage-gated potassium channels. Using a sophisticated set of experiments the authors conclude that the maximum output per individual docking site allows for estimating the occupancy of the docking site under resting conditions.

**Decision letter after peer review:**

Thank you for submitting your article "Incomplete vesicular docking limits synaptic strength under high release probability conditions" for consideration by *eLife*. Your article has been reviewed by Eve Marder as the Senior Editor, a Reviewing Editor, and three reviewers. The reviewers have opted to remain anonymous.

The reviewers have discussed the reviews with one another, and the Reviewing Editor has drafted this decision to help you prepare a revised submission.

Summary:

In this manuscript, the authors continue their work aiming at a better understanding of the readily releasable pool of vesicles. To this end, they use a preparation allowing recording from a single-site synapse. This approach allows for counting synaptic vesicle fusion events rather than relying on the EPSC for mean-variance analyses. From this analysis, they estimate the number of release sites, and the release probability per site. By increasing the release probability of docked vesicles, they are able to estimate the fraction of release sites with docked vesicles. The work is a continuation of an already rather extensive body of data provided by the Marty group. Here they use different calcium concentrations and TEA in order to drive release probability to 1, which allows them to obtain a better estimate of the occupancy of vesicle docking sites.

Essential revisions:

All reviewers agree that the manuscript is interesting, experimentally well done, and allows for further insights into the physiological foundation of the classical synaptic release parameters, thus deserving publication in *eLife* after appropriate revision. There was some disagreement between the reviewers concerning the use of TEA for blocking K-channels. After some discussion, the reviewers agreed on suggesting one set of additional experiments to strengthen the manuscript: To ensure that calcium influx is adequate to fuse all docked vesicles, an experiment should be carried out in which TEA is complemented with moderate concentrations of 4-AP (in the range of 10-20 uM). If, as expected, probability of fusion = 1.0 with TEA alone, further increases in calcium by blocking additional K-channels with 4-AP is not expected to change the response. Such an experiment would give more confidence that under the TEA conditions indeed maximal calcium concentrations are achieved as claimed.

*Reviewer #1:*

1) Authors assume that both p and δ are Ca^2+^-dependent but argue that Ca^2+^ influx increased by K channel blocker (TEA increase from 1mM to 5 mM) has no effect on δ. The references quoted (subsection “TEA and [Ca^2+^]_o_ increase P by two different mechanisms”) do not seem to provide a support this argument. At least a spatial model to distinguish Ca^2+^ increases, by elevation of external Ca^2+^ concentration and by TEA, would be necessary to propose this hypothesis.

2) In subsection “Saturation of P as a function of TEA concentration”, P1 values in 1 mM TEA and 5 mM TEA are compared, but Figure 1E indicates that control values (1.5 Ca^2+^, 0 TEA) are very different between two sets of experiments, therefore this comparison does not make sense unless the ratio to controls in individual experiments are compared. Likewise, at in subsection “Combining measurements of Ca2+ 334 entry with P measurements”, Ca^2+^ influx is compared between two conditions to be similar (1 mM TEA / 3 mM Ca^2+^ (green in Figure 3C) vs 5 mM TEA / 1.5 mM Ca^2+^ (purple), with no controls across two sets of experiments.

Reviewer #2:

The experiments are convincing, and the data provide a simple explanation for complex behaviors of synapses. For those interested in molecular mechanisms, the manuscript spins out a multitude of questions about how to square their model with ultrastructural data, and molecular models of docking and of modulation, including the roles of Doc2, synaptotagmins, and Munc13 proteins. These are largely unaddressed in the Discussion. But I feel it is better to keep the manuscript narrow than becoming a venue for speculation.

My critique mainly focuses on presentation.

1) Rationale. It is still not clear what conclusions had been made in previous manuscripts, and what were the shortcomings of these publications that this manuscript corrected. At first it seems like well-trod ground for the Marty lab – docking is limiting, increases in docking underlie facilitation. In the end, I had to reread all of the previous manuscripts to determine what had been learned here. In fact, docking values were slightly more (δ=0.45 Miki, 2016; δ=0.3 Miki, 2018) – so what has been learned? They should state succinctly what was previously known, describe the limitations in these previous manuscripts, specifically why it was difficult to precisely determine the docking site occupancy in their previous studies. And underscore the novel result, that is, that calcium increases the number of docked vesicles even before stimulation.

2) Abstract. The authors do not attempt to reach out to the general *eLife* reader, even in the Abstract – the significant take-home messages are lost in comments about jitter and saturation. The general reader will not understand the mode of action or the significance of TEA treatment. I appreciate the specific numbers; the mean N value should also be cited.

3) Acronyms. In the rest of the document, the English style is simple and direct, which is an achievement in itself. Acronyms such as 'AP', 'MLI', 'PF', and 'PPR', are not defined. I think the reader will also need an occasional reminder to distinguish 'P' from 'p' from "release probability" as they read on.

4) TEA. The conclusion that docking is limiting hangs on the argument that they have maximized p with 1mM TEA – 5mM TEA does not increase P1. But that claim is not fully supported. The alternative is that 1mM TEA saturates calcium influx. The experiment demonstrating that TEA has not saturated calcium influx does not come until Figure 3. In the meantime, I was convinced they were wrong. Rearranging the figures might improve the logic of the argument or noting that this will be demonstrated below. Moreover, the ΔF/F_o_ data for 1mM TEA & 1.5 mM Ca^2+^ varied dramatically between two comparisons in Figure 3. They were more convincingly demonstrated in the 2016 paper. Nevertheless, they achieve significance and the data are consistent with previous experiments; they can stand in my opinion.

5) Probability of release. subsection “Effect of adding 1 mM TEA on P”: What are the estimated p values for these values of P1? It would useful to include the SV count, δ and p1 that went into these values for {plus minus} TEA. I managed to screw up my math from my back-of-the-envelope calculations when I was checking those values from the data.

6) Calcium channels. What does p=1 mean for our models of the active zone? These data suggest that every docked vesicle can be released. That suggests:

- All SVs docked near calcium channels and all calcium channels open, or

- Calcium domains are as large as the active zone and access all SVs.

7) Replenishment. Won't 'crash fusions' from the replacement pool (described in Mike et al., 2018) interfere with their SV count after the first stimulation in the train. These will be included in their occupancy numbers.

*Reviewer #3:*

1) I would guess that changing bath calcium alters resting intracellular calcium, thus affecting δ. Did the authors' calcium measurements show such changes?

2) One might imagine that TEA could depolarize the terminal and also lead to a change in calcium, yet the authors believe there is no effect of TEA on δ. Some comment about the effect of TEA is needed to help the reader understand its singular action on the spike waveform.

3) In subsection “Combining measurements of Ca^2+^ entry with P measurements”, highly sensitive TO Ca.

4) Subsection “Comparison with other preparations” 'notable exception' of Silver's paper. What is 'normal [Ca^2+^]_o_' that gave a P of 0.9, given that physiological Ca/Mg levels were not used in that work?

5) Are there data on how 1 and 5 mM TEA affect the spike in granule cells?

---

## [Author Response]

Essential revisions:All reviewers agree that the manuscript is interesting, experimentally well done, and allows for further insights into the physiological foundation of the classical synaptic release parameters, thus deserving publication in eLife after appropriate revision. There was some disagreement between the reviewers concerning the use of TEA for blocking K-channels. After some discussion, the reviewers agreed on suggesting one set of additional experiments to strengthen the manuscript: To ensure that calcium influx is adequate to fuse all docked vesicles, an experiment should be carried out in which TEA is complemented with moderate concentrations of 4-AP (in the range of 10-20 uM). If, as expected, probability of fusion = 1.0 with TEA alone, further increases in calcium by blocking additional K-channels with 4-AP is not expected to change the response. Such an experiment would give more confidence that under the TEA conditions indeed maximal calcium concentrations are achieved as claimed.

To address this point, we have performed experiments starting in 1.5 mM external calcium and 1 mM TEA, adding 15 µM 4-AP. Group results are shown in Figure 4—figure supplement 1 of the revised manuscript. Mean P1 values were 0.140 in TEA, and 0.187 in TEA + 4-AP. In TEA + 4-AP, both facilitation and 2-step release were surprisingly prominent, indicating very strong calcium entry. After correcting for 2-step release, the P1 value in TEA + 4-AP was 0.125, similar to the value in TEA alone. Therefore, the new data are in conformity with the prediction outlined by the Editor, that when starting in TEA, further addition of 4-AP should not increase P1 significantly. Put in other words, the new results suggest that 4-AP, just like TEA, acts on p but not on δ.

Reviewer #1:

*1) Authors assume that both p and δ are Ca^2+^-dependent but argue that Ca^2+^ influx increased by K channel blocker (TEA increase from 1mM to 5 mM) has no effect on δ. The references quoted (subsection “*TEA and [Ca^2+^]_o_ increase P by two different mechanisms”*) do not seem to provide a support this argument. At least a spatial model to distinguish Ca^2+^ increases, by elevation of external Ca^2+^ concentration and by TEA, would be necessary to propose this hypothesis.*

The reference Pulido, 2018 helps explaining that, while increasing [Ca^2+^]_o_ to 3 mM and adding 1 mM TEA have similar effects on P1, the first manipulation severely decreases facilitation but the second does not. As shown in Pulido, 2018, docking site models predict that short term synaptic plasticity does not only depend on p, but also on δ, as well as on r, the probability of replenishment of an emptied docking site during one interpulse interval. Specifically, PPR = (1 – p) + pr + (1/ δ – 1) r (eq 5 in Pulido, 2018). As can be seen from this equation, when p increases to its maximum of p = 1, the PPR decreases to a minimum of r/ δ, which is larger than 1 (facilitating) if r > δ. Therefore, certain synapses with low δ values remain facilitating even if p = 1. When on the other hand δ increases to its maximum of 1, the PPR decreases to 1 – p + pr, which is always smaller than 1 (as both p and r are smaller than 1). Therefore, all synapses become depressing if δ increases to its maximum of δ = 1. We have changed the text to make the argument clearer.

The new data with 4-AP in addition to TEA shows that in these conditions, P1 remains low while facilitation remains prominent (new Figure 4—figure supplement 1). This confirms that blocking voltage dependent K channels, either with TEA or with 4-AP, has different effects on synaptic function than increasing [Ca^2+^]_o_.

To address the issue of differences in mechanism, we have now included new data contrasting the effects of elevated [Ca^2+^]_o_ and adding TEA on miniature current frequencies. These experiments show that elevated [Ca^2+^]_o_ increases mini frequency, but that addition of TEA does not have any such effect. They are consistent with our suggestion that the basal presynaptic Cai is increased following the first maneuver, but not following the second one. Also, the new experiments mixing TEA and 4-AP (see above, response to Editor) gave results consistent with the hypothesis that TEA (or other K channel blockers) act on p, not on δ.

2) In subsection “Saturation of P as a function of TEA concentration”, P1 values in 1 mM TEA and 5 mM TEA are compared, but Figure 1E indicates that control values (1.5 Ca^2+^, 0 TEA) are very different between two sets of experiments, therefore this comparison does not make sense unless the ratio to controls in individual experiments are compared. Likewise, at in subsection “Combining measurements of Ca2+ 334 entry with P measurements”, Ca^2+^ influx is compared between two conditions to be similar (1 mM TEA / 3 mM Ca^2+^ (green in Figure 3C) vs 5 mM TEA / 1.5 mM Ca^2+^ (purple), with no controls across two sets of experiments.

The reviewer is right: a comparison of P1 between 1 mM TEA and 5 mM TEA is only indicative with the present data because the experiments were separate. To acknowledge this, we have changed our text in subsection “Combining measurements of Ca^2+^376 entry with P measurements” to say that our results were only an indication that P1 had similar values in 1 mM TEA and in 5 mM TEA. The important point is that both in 1 mM TEA and in 5 mM TEA, P1 is far from its maximum value of 1. Likewise, we have attenuated the statement concerning 1 mM TEA / 3 mM Ca^2+^ and 5 mM TEA / 1.5 mM Ca^2+^ in subsection “Combining measurements of Ca^2+^ entry with P measurements”. In Figure 4C, the various data points are presented with their respective error bars, and the scatter of experimental data is plainly apparent. In this figure all available results have been incorporated, so that means are slightly different from the means presented in Figure 1, that only reported the experiments with paired analysis. Therefore in Figure 4C, the n value is 9 for 1.5 Ca + 1 TEA, not 5, and the n value is 6 for 1.5 Ca + 5 TEA, not 4. Incorporating more data in the summary plot of Figure 4C gives more statistical weight to mean values. The point here is that the data cannot be fit with a single scaled version of the model curve in Figure 4B.

We have performed the ratio analysis suggested by the reviewer. The mean P1 ratio is 1.9 in 1 TEA, and 3.0 in 5 TEA, but the difference is not significant. We have incorporated this comparison in the revised in subsection “Saturation of P as a function of TEA concentration”.

Reviewer #2:[…]1) Rationale. It is still not clear what conclusions had been made in previous manuscripts, and what were the shortcomings of these publications that this manuscript corrected. At first it seems like well-trod ground for the Marty lab – docking is limiting, increases in docking underlie facilitation. In the end, I had to reread all of the previous manuscripts to determine what had been learned here. In fact, docking values were slightly more (δ=0.45 Miki 2016; δ=0.3 Miki, 2018) – so what has been learned? They should state succinctly what was previously known, describe the limitations in these previous manuscripts, specifically why it was difficult to precisely determine the docking site occupancy in their previous studies. And underscore the novel result, that is, that calcium increases the number of docked vesicles even before stimulation.

We have rewritten the end of Introduction along the lines indicated by the reviewer. In addition, we have added a sentence to the first paragraph of Discussion section to highlight the advances achieved by the paper.

2) Abstract. The authors do not attempt to reach out to the general eLife reader, even in the Abstract – the significant take-home messages are lost in comments about jitter and saturation. The general reader will not understand the mode of action or the significance of TEA treatment. I appreciate the specific numbers; the mean N value should also be cited.

We have rewritten the Abstract along the lines indicated by the reviewer.

3) Acronyms. In the rest of the document, the English style is simple and direct, which is an achievement in itself. Acronyms such as 'AP', 'MLI', 'PF', and 'PPR', are not defined. I think the reader will also need an occasional reminder to distinguish 'P' from 'p' from "release probability" as they read on.

Done.

4) TEA. The conclusion that docking is limiting hangs on the argument that they have maximized p with 1mM TEA – 5mM TEA does not increase P1. But that claim is not fully supported. The alternative is that 1mM TEA saturates calcium influx. The experiment demonstrating that TEA has not saturated calcium influx does not come until Figure 3. In the meantime, I was convinced they were wrong. Rearranging the figures might improve the logic of the argument or noting that this will be demonstrated below. Moreover, the ΔF/F_o_ data for 1mM TEA & 1.5 mM Ca^2+^ varied dramatically between two comparisons in Figure 3. They were more convincingly demonstrated in the 2016 paper. Nevertheless, they achieve significance and the data are consistent with previous experiments; they can stand in my opinion.

The new experiments combining TEA and 4-AP (see above response to Editor, and new Figure 4—figure supplement 1) indicate that, no matter the method used to increase calcium entry, P1 remains limited by δ. These new data reinforce our suggestion that TEA acts on p, not on δ.

5) Probability of release. subsection “Effect of adding 1 mM TEA on P”: What are the estimated p values for these values of P1? It would useful to include the SV count, δ and p1 that went into these values for {plus minus} TEA. I managed to screw up my math from my back-of-the-envelope calculations when I was checking those values from the data.

P1 was 0.213 in 1.5 calcium and 1 TEA, while δ is estimated at 0.22 from Figure 4 under these conditions. By inverting the equation P1 = δ p, one obtains p = 0.97. However, presenting this calculation together with the value of P1 on p. 4 would be premature since the estimate of δ appears only later in the manuscript, as a result of the analysis of Figure 4C. At the present stage P1 is simply obtained by making the ratio of n_1_ over N.

The results of p calculations in TEA have now been incorporated in subsection "Correcting release probability for docking site occupancy".

6. Calcium channels. What does p=1 mean for our models of the active zone? These data suggest that every docked vesicle can be released. That suggests:- All SVs docked near calcium channels and all calcium channels open, or- Calcium domains are as large as the active zone and access all SVs.

We now state (subsection “Effect of adding 1 mM TEA on P”) that we assume all sites to have the same sensitivity to calcium, and to be located at similar distances from calcium sources. This is in conformity with our previous work (Miki et al., 2016; 2017). Because PF-MLI AZs are very small and have several clusters of VGCCs, it is likely that individual SVs are close to several VGCCs (Miki et al., 2017).

7) Replenishment. Won't 'crash fusions' from the replacement pool (described in Mike et al., 2018) interfere with their SV count after the first stimulation in the train. These will be included in their occupancy numbers.

Fortunately, 2-step release in response to the 1st AP is negligible in most conditions. Only later in a train does 2-step release appear. As our assessment of δ values only relies on an analysis of P1, errors due to 2-step release are minimal. According to results exposed in Miki et al., 2018, 2-step release following the first AP becomes detectable only in the most extreme conditions (3 mM calcium and 1 mM TEA, or 1.5 mM calcium, 1 mM TEA and 4-AP, see above). This effect is accounted for as explained in in subsection “Combining measurements of Ca^2+^376 entry with P measurements”, by removing 2-step release events from the relevant points in the plot of Figure 4C. We have expanded the text explaining the correction to clarify this issue.

Reviewer #3:1) I would guess that changing bath calcium alters resting intracellular calcium, thus affecting δ. Did the authors' calcium measurements show such changes?

In principle the reviewer is right. Unfortunately, our calcium measurements do not address this issue. In individual experiments we compared a control situation (either 1.5 or 3 external calcium) without TEA to a test situation with TEA (or in one series of experiments, we increased the TEA concentration). Thus, we cannot conclude from these experiments whether changing calcium would increase the basal calcium level. However, we do have evidence indicating an increase of intracellular calcium, from measurements of mini rates at the two external calcium concentrations. We find that the mini rate increases as a function of external calcium, suggesting a rise in internal calcium as well. This information has now been included in the revised manuscript.

2) One might imagine that TEA could depolarize the terminal and also lead to a change in calcium, yet the authors believe there is no effect of TEA on δ. Some comment about the effect of TEA is needed to help the reader understand its singular action on the spike waveform.

Contrary to the above results with 3 mM calcium, adding TEA does not change miniature current frequency, indicating no change in basal presynaptic calcium. We have now added this information to the Results section.

3) In subsection “Combining measurements of Ca^2+^ entry with P measurements”, highly sensitive TO Ca.

Fixed.

4) Subsection “Comparison with other preparations” 'notable exception' of Silver's paper. What is 'normal [Ca^2+^]_o_' that gave a P of 0.9, given that physiological Ca/Mg levels were not used in that work?

Indeed, several external concentrations of calcium and magnesium were used in Silver's work. When looking at Figure 9A of this paper however, it seems that in 2 mM calcium and 1 mM magnesium, the parabola is almost complete, indicating P near 0.9. Or did we misunderstand this figure?

5) Are there data on how 1 and 5 mM TEA affect the spike in granule cells?

We have added a figure (Figure 1—figure supplement 1) to illustrate the effects of 1 mM TEA on spikes.